# OLoRA+: A Hybrid Approach to Parameter-Efficient Fine-Tuning of Large Language Models

**S.I.Kushmuratov, F.T. Adilova, R.R. Davronov**
Laboratory of Biomedical Informatics
V.I.Romanovskiy Institute of Mathematics, Uzbekistan Academy of Sciences
9 University Street, Tashkent 100174, Uzbekistan
bekmezonali@gmail.com, fatadilova@mathinst.uz, rifqat@gmail.com

## Abstract

Parameter-Efficient Fine-Tuning (PEFT) is essential for adapting Large Language Models (LLMs) under resource constraints, yet existing methods often treat initialization and optimization as separate concerns. This paper introduces OLoRA+, a novel hybrid approach that synergistically combines the structural stability of Orthonormal Low-Rank Adaptation (OLoRA) with the accelerated convergence of LoRA+. By initializing adapter matrices via QR decomposition of pre-trained weights and applying differential learning rates to the upstream and downstream projection matrices, OLoRA+ aims to enhance both stability and feature learning speed. We evaluated the method on the LLMs models using a subset of the Alpaca instruction-following dataset. Empirical results demonstrate that OLoRA+ consistently outperforms the standard OLoRA baseline across Evaluation Loss, BLEU, and ROUGE metrics without incurring additional computational costs. Crucially important that our analysis uncovers two distinct effective learning regimes: a "Refinement" strategy (learning rate ratio $\lambda < 1$) that optimizes the initial orthonormal basis, and an "Exploration" strategy ($\lambda > 1$) that seeks new parameter directions. These findings suggest that OLoRA+ offers a more versatile and robust framework for efficient LLM adaptation than its predecessors.

**Keywords:** LLM, Fine-tuning, PEFT, Low-Rank Adaptation, QR Decomposition, Differential Optimization.

## 1 Introduction

The advent of pre-trained large language models (LLMs) has significantly advanced the field of natural language processing (NLP) (Bommasani et al., 2021). These models demonstrate powerful capabilities in general language understanding and generation. However, the increasing size of LLMs presents significant challenges for traditional full fine-tuning, which updates all model parameters and incurs substantial computational and memory costs (Devlin et al., 2018).

In response to these challenges, Parameter-Efficient Fine-Tuning (PEFT) methods have emerged as a promising solution. PEFT techniques adapt pre-trained models by selectively fine-tuning a small number of additional parameters while keeping the majority of the original model frozen. Among these, Low-Rank Adaptation (LoRA) (Hu et al., 2021) has become one of the most popular approaches, achieving strong performance without increasing inference latency. The success of LoRA has spurred the development of an ecosystem of variants, each aimed at addressing specific limitations of the original method.

Research into improving LoRA has progressed along several distinct directions. On one hand, methods like OLoRA have focused on the initialization of the adapter (Büyükakyüz, 2024). OLoRA utilizes QR decomposition to create an orthonormal basis from the pre-trained weights, providing a more stable and well-conditioned starting point for training, which leads to accelerated convergence. On the other hand, approaches like LoRA+ have focused on the optimization of the training process.

LoRA+ demonstrates that using differential learning rates for the two adapter matrices ($A$ and $B$) can significantly accelerate training and improve final performance (Hayou et al., 2024).

However, these research directions—improving initialization and optimizing the training process—have largely evolved in parallel. The potential synergistic effects of their combination remain an unexplored area. This paper addresses this gap by introducing OLoRA+, a novel hybrid method that synergistically combines the structured initialization of OLoRA with the accelerated optimization of LoRA+.

The primary goal of this research is to investigate the interaction between these two techniques and determine if their combination yields a more effective and versatile PEFT method. Through empirical evaluation, we not only compare the performance of OLoRA+ against the OLoRA baseline but also uncover a new learning dynamic related to the choice of the learning rate ratio. We characterize these regimes as "Refinement" and "Exploration" strategies, demonstrating that OLoRA+ surpasses its predecessors and offers a deeper understanding of the interplay between adapter initialization and optimization.

## 2 RELATED WORK

Our work builds directly upon three foundational PEFT methods: LoRA (Hu et al., 2021), OLoRA (Büyükakyüz, 2024), and LoRA+ (Hayou et al., 2024).

### 2.1 LOW-RANK ADAPTATION (LoRA)

LoRA is based on the hypothesis that the change in a model's weights during adaptation occurs in a "low-dimensional space". It freezes the pre-trained model weights and injects a pair of trainable low-rank matrices, $A$ and $B$, into each target layer which is shown in Figure 1. The update is represented as $\Delta W = BA$.

$$W = W_0 + \Delta W = W_0 + BA \tag{1}$$

To ensure a non-disruptive start to training, matrix $B$ is initialized to zeros, making the initial update zero. While highly parameter-efficient, LoRA can converge more slowly than full fine-tuning (Hu et al., 2021).

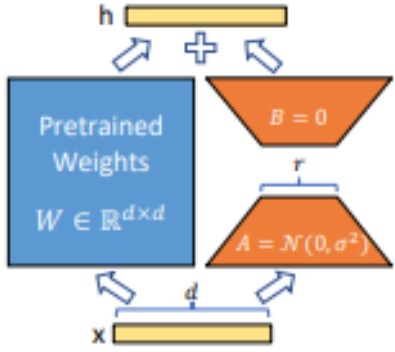

Figure 1: Reparametrization of weights for train only A and B

### 2.2 ORTHONORMAL LoRA (OLoRA)

OLoRA addresses the convergence speed and stability limitations of LoRA by improving the initialization process. Instead of random initialization, OLoRA performs a QR decomposition on the pre-trained weight matrix $W$ to obtain an orthogonal matrix $Q$ and an upper-triangular matrix $R$ (Büyükakyüz, 2024). The adapter matrices $B$ and $A$ are then initialized with the first $r$ columns of

$Q$ and $r$ rows of $R$ (Aghajanyan et al., 2020; Meng et al., 2024), respectively, which is shown in Figure 2.

$$W_{\text{adapted}} = W + Q_r R_r \tag{2}$$

This orthonormal initialization provides a "well-conditioned optimization landscape," leading to faster convergence and often better final performance compared to standard LoRA.

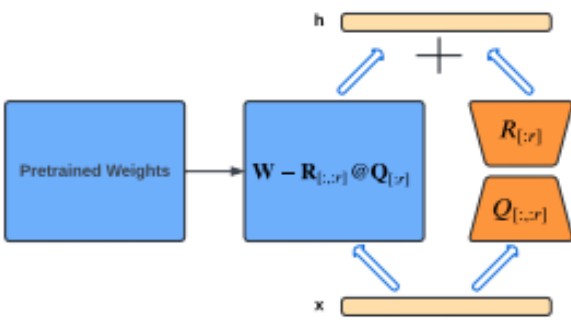

Figure 2: Illustration of the OLoRA method

## 2.3 LoRA+

LoRA+ identifies a suboptimality in the standard LoRA training procedure, where both adapter matrices $A$ and $B$ are updated with the same learning rate. The authors (Hayou et al., 2024) argue that this slows down feature learning. LoRA+ corrects this by setting a much higher learning rate for matrix $B$ than for matrix $A$ (i.e., $\eta_B = \lambda \eta_A$, where $\lambda \gg 1$). This simple change is shown to improve performance by 1-2% and accelerate training speed by up to 2x at no additional computational cost.

$$A \leftarrow A - \eta \times G_A, \quad B \leftarrow B - \lambda \eta \times G_B, \quad \lambda \gg 1 \tag{3}$$

## 3 THE OLoRA+ METHOD

Our central hypothesis is that the benefits of OLoRA's superior initialization and LoRA+'s accelerated optimization are complementary and can be combined to create a more powerful PEFT method.

### 3.1 ARCHITECTURAL SYNERGY

OLoRA+ is a hybrid method that integrates these two techniques in a two-phase conceptual process: a structured initialization followed by a differential optimization.

The process begins with the **OLoRA initialization phase**. For a given pre-trained weight matrix $W$, we first perform a QR decomposition, $W = QR$. The adapter matrices, denoted as $B$ (up-projection) and $A$ (down-projection), are then initialized using the results of this decomposition. Specifically, $B$ is initialized with $Q_r$ (the first $r$ columns of the orthogonal matrix $Q$) and $A$ is initialized with $R_r$ (the first $r$ rows of the upper-triangular matrix $R$). This provides a structured, orthonormal starting point that is a low-rank approximation of the original weights, creating a "well-conditioned optimization landscape" (Büyükakyüz, 2024). The adapted weight matrix is thus expressed as $W_{\text{adapted}} = W + BA$, where $B$ and $A$ start as $Q_r$ and $R_r$.

$$W = W_0 + \Delta W = W_0 + BA = W_0 + Q_r R_r \tag{4}$$

Once training begins, the **LoRA+ optimization phase** is applied. Instead of using a single learning rate $\eta$ for both adapter matrices, we introduce a learning rate ratio, $\lambda$. The update rules for the matrices during each step of gradient descent are governed by the following formulas:

$$A \leftarrow A - \eta \times G_A \tag{5}$$
$$B \leftarrow B - \lambda\eta \times G_B \tag{6}$$

Here, $G_A$ and $G_B$ represent the gradients for matrices $A$ and $B$, respectively. The base learning rate is $\eta$, and the effective learning rate for matrix $B$ is scaled by the ratio $\lambda$. The LoRA+ paper recommends $\lambda \gg 1$ for standard LoRA to accelerate feature learning. This synergy—starting from a superior initial state and then following a more efficient learning path—is the core of the OLoRA+ method.

## 3.2 IMPLEMENTATION DETAILS

We implemented OLoRA+ by leveraging the Hugging Face PEFT library.

- **OLoRA Initialization:** We configured the `LoraConfig` by setting the `init_lora_weights="olora"` parameter. This instructs the library to perform the QR decomposition and initialize the $A$ and $B$ matrices accordingly.

- **LoRA+ Optimization:** We created a custom optimizer by separating the model's trainable parameters into two groups based on their names ($A$ and $B$). We then instantiated an AdamW optimizer, assigning different learning rates to each group based on a specified $\lambda$. This custom optimizer was then passed to the Hugging Face Trainer.

## 4 EXPERIMENTAL SETUP

To validate our method, we conducted a controlled experiment comparing OLoRA+ against a standard OLoRA (Büyükakyüz, 2024) baseline.

### 4.1 MODELS

All experiments were conducted using the TinyLlama-1.1B, OPT-1.3B and Gemma-2B models, compact yet capable LLMs suitable for efficient experimentation (Zhang et al., 2024).

### 4.2 DATASET

We used a subset of the tatsu-lab/alpaca dataset, which consists of 52,000 instruction-following demonstrations generated by OpenAI's text-davinci-003 engine (Taori et al., 2023). For our experiments, we used a split of 1000 samples for training and 500 samples for evaluation to simulate a resource-constrained fine-tuning scenario.

### 4.3 BASELINES AND HYPERPARAMETERS

To ensure a fair comparison, all key hyperparameters were kept consistent across experiments:

- **Rank (r):** 32
- **Alpha ($\alpha$):** 16
- **Learning Rate:** 3e-4
- **Epochs:** 1
- **Baseline (OLoRA):** The model was trained using the OLoRA initialization with a standard AdamW optimizer (Loshchilov & Hutter, 2017).
- **Our Method (OLoRA+):** The model was trained using the OLoRA initialization and our custom LoRA+ optimizer with a varying learning rate ratio.

## 4.4 EVALUATION METRICS

Model performance was assessed using a suite of standard metrics:

- **Evaluation Loss:** To measure the model's ability to generalize to unseen data during training (Wikipedia contributors, 2025b).
- **BLEU:** To measure the precision and fluency of the generated text (Wikipedia contributors, 2025a).
- **ROUGE:** Specifically ROUGE-1, ROUGE-2, and ROUGE-L, to measure the recall of key information in the generated text (Wikipedia contributors, 2025c).

## 5 RESULTS AND DISCUSSION

Our experiments confirmed our initial hypothesis and led to a significant new discovery regarding the learning dynamics of initialized adapters.

### 5.1 PERFORMANCE IMPROVEMENT

Across all quantitative metrics, our OLoRA+ method consistently outperformed the OLoRA baseline. The model trained with OLoRA+ achieved a lower final evaluation loss and higher BLEU and ROUGE scores, indicating superior generalization and text generation quality which is shown in Tables 1, 2, and 3.

Table 1: Evaluation results by TinyLlama-1.1B

| Method | Eval Loss | ROUGE-1 | ROUGE-2 | ROUGE-L | BLEU |
|---|---|---|---|---|---|
| OLoRA (Baseline) | 88.49 | 74.53 | 56.46 | 71.78 | 56.81 |
| OLoRA+ ($\lambda \ll 1$) | 88.22 | 74.56 | 56.41 | 71.86 | 56.83 |
| OLoRA+ ($\lambda \gg 1$) | 88.39 | 74.52 | 56.49 | 71.86 | 56.83 |

Table 2: Evaluation results by OPT-1.3B

| Method | Eval Loss | ROUGE-1 | ROUGE-2 | ROUGE-L | BLEU |
|---|---|---|---|---|---|
| OLoRA (Baseline) | 75.72 | 63.53 | 62.32 | 73.08 | 68.45 |
| OLoRA+ ($\lambda \ll 1$) | 75.68 | 62.56 | 62.38 | 73.13 | 68.56 |
| OLoRA+ ($\lambda \gg 1$) | 75.41 | 64.52 | 62.33 | 73.25 | 68.53 |

Table 3: Evaluation results by Gemma-2B

| Method | Eval Loss | ROUGE-1 | ROUGE-2 | ROUGE-L | BLEU |
|---|---|---|---|---|---|
| OLoRA (Baseline) | 81.22 | 83.78 | 59.25 | 78.56 | 62.14 |
| OLoRA+ ($\lambda \ll 1$) | 81.18 | 83.82 | 59.36 | 78.59 | 62.19 |
| OLoRA+ ($\lambda \gg 1$) | 81.13 | 83.81 | 59.33 | 78.61 | 62.19 |

As shown in Figure 3 and Table 4, our proposed OLoRA+ method demonstrates clear superiority across all metrics. According to (a), Tiny-llama demonstrated significantly lower testing losses than the other LLMs. Graphs (b), (c), (d), and (e) show the performance for the BLEU and ROUGE text generation metrics. In each case, both OPT-1.3 and Gemma-2B LLM outperform or equal the baseline.

### 5.2 THE DISCOVERY OF DUAL LEARNING REGIMES

The most significant finding of our research is that, contrary to the recommendations in the original LoRA+ paper, OLoRA+ achieves strong performance with a learning rate ratio ($\lambda$) both greater

Table 4: Evaluation results by LLMs

| LLMs | Eval Loss | ROUGE-1 | ROUGE-2 | ROUGE-L | BLEU |
|------|-----------|---------|---------|---------|------|
| Tiny-llama | 88.22 | 74.56 | 56.49 | 71.86 | 56.83 |
| OPT-1.3B | 75.41 | 64.52 | 62.38 | 73.25 | 68.56 |
| Gemma-2B | 81.13 | 83.82 | 59.36 | 78.61 | 62.19 |

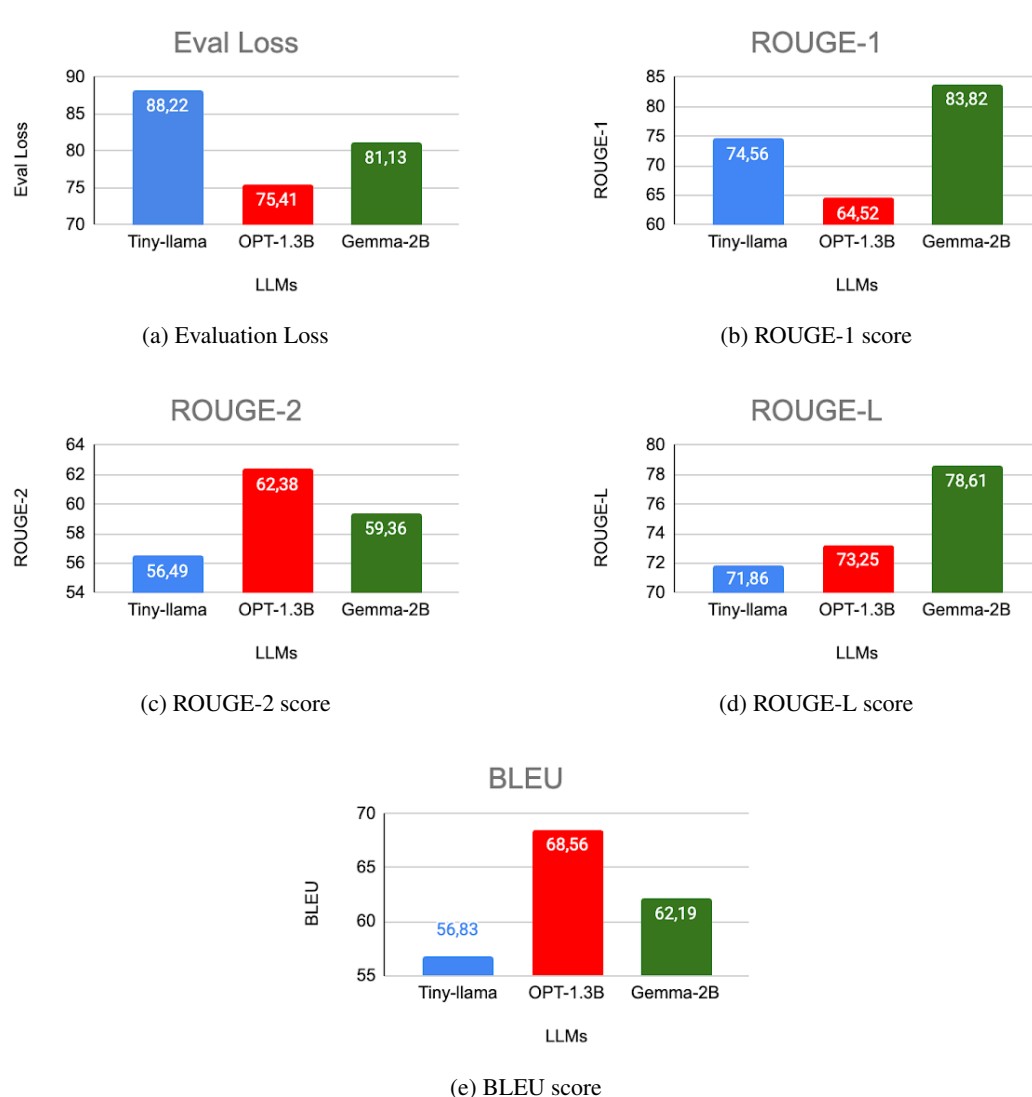

(a) Evaluation Loss

(b) ROUGE-1 score

(c) ROUGE-2 score

(d) ROUGE-L score

(e) BLEU score

Figure 3: Comparative analysis of the performance of metrics by LLMs.

than 1 and less than 1. This discovery suggests that the optimal optimization strategy is contingent on the initialization method. We characterize these two effective regimes as a trade-off between "Refinement" and "Exploration."

### 5.3 CHARACTERIZING "REFINEMENT" VS. "EXPLORATION"

**The Refinement Strategy** ($\lambda < 1$)**:** When the learning rate for matrix $A$ (derived from $R_r$) is higher than for matrix $B$ (derived from $Q_r$), the model adopts a conservative strategy. It trusts the high-quality orthonormal directions provided by the OLoRA initialization and focuses the learning

effort on finding the correct magnitudes and combinations for these directions. This approach gently refines the strong initial structure. Our results show this is a highly effective strategy, suggesting that for many tasks, the primary challenge is learning *how* to use the initial basis, not finding a new one.

**The Exploration Strategy ($\lambda > 1$):** When the learning rate for matrix $B$ is higher than for matrix $A$, the model adopts a more aggressive strategy. It uses the OLoRA initialization as a launchpad but actively explores for new directions in the parameter space by more rapidly updating the $B$ matrix. This approach is more willing to deviate from the initial orthonormal structure in search of a potentially better solution. This aligns with the original LoRA+ logic and is effective for tasks that may require adaptation beyond the initial subspace.

## 6 CONCLUSION

In this paper, we introduced OLoRA+, a novel hybrid PEFT method that synergistically combines the orthonormal initialization of OLoRA with the differential optimization of LoRA+. Results of computational experiments demonstrate that OLoRA+ consistently outperforms the OLoRA baseline in instruction-following tasks, achieving lower evaluation loss and higher BLEU and ROUGE scores via different LLMs which are TinyLlama-1.1B, OPT-1.3B and Gemma-2B without any additional computational cost.

The key novelty of this study is the discovery and characterization of two distinct and effective learning regimes for OLoRA+: a "Refinement" strategy (ratio $< 1$) and an "Exploration" strategy (ratio $> 1$). This finding reveals that the optimal optimization strategy is fundamentally dependent on the initialization scheme, offering a new dimension for hyperparameter tuning. This research not only presents OLoRA+ as a more versatile and powerful approach for LLM adaptation but also provides a deeper understanding of the critical interplay between adapter initialization and optimization dynamics, paving the way for more effective fine-tuning in resource-constrained environments.

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
