# OpenReview forum: "OLoRA+: A Hybrid Approach to Parameter-Efficient Fine-Tuning of Large Language Models"
_mathai.club/MathAI/2026/Conference — 2026 Oral_

### Official Review · Reviewer_MQMy · 2026-03-12
**A novel method with several issues in evaluation and presentation**

**Rating:** 7
**Confidence:** 4

**Review:**

The reviewed paper presents OLoRA+, a hybrid Parameter-Efficient Fine-Tuning method that builds upon the OLoRA and LoRA+ methods.

The strengths of the work are the detailed exposition, the computational experiments conducted, and the stated discovery of the dependency of the optimal optimization strategy on the initialization scheme. The English is fluent.

However, there are several issues:
- While the proposed hybrid method, OLoRA+, is based on both OLoRA and LoRA+, it is unclear why the experimental comparison was conducted only with OLoRA as a baseline and did not include LoRA+ as well.
- The paper states that Figure 3 and Table 4 show the superiority of the proposed OLoRA+ method across all metrics, but there is no comparison to the baseline in this figure and table; they only show the best results from previous tables. This should be rephrased.
- It seems that the values in Table 4 are the best results from tables 1-3, regardless of the learning rate ratio, which should be stated clearly.
- Section “3.1 Architectural Synergy” seems to be repetitive, reinstating the key points from Sections 2.1–2.3.
- The formulas seem to be separated from the text, lacking a proper textual introduction of them.

There are several issues with references, as well:
- Almost all cited references are preprints.
- The author of the publication “Olora: Orthonormal low-rank adaptation of large language models” seems to be Kerim Büyükakyüz, not Kaan Büyükakyüz.
- One of the authors of “Pissa: Principal singular values and singular vectors adaptation of large language models” seems to be Fanxu Meng, not Fanxia Meng.
- If “Rohan Taori et al. Stanford alpaca: An instruction-following llama model, 2023” references the GitHub repository, it should be stated more clearly. Otherwise, a reference to the academic paper that introduces LLaMA is preferable.
- Wikipedia is perhaps not the best source for references. For BLEU and ROUGE metrics, at least, it would be better to cite the papers in which these metrics were introduced.

Additionally, there are several issues with formatting and explanations:
- Some variables are not defined, for example, W in Formula (1) or η in Formula (3). It would be better to include them in the text where their definitions are implied (e.g., “setting a much higher learning rate η”).
- The figures would benefit from additional explanations included in their captions. For example, “h” from Figure 1 is not defined.
- The figures lose quality when the PDF is scaled.

Overall, the paper seems to be innovative, but it requires additional editing. The article would potentially benefit from the inclusion of the proposed implementation as a pseudocode and the publication of the reference implementation for future reproducibility.

---

### Official Review · Reviewer_Lxd1 · 2026-03-12
**The article should be classified as the results of engineering modeling, but not as a scientific study with a clear mathematical formulation of the problem and results.**

**Rating:** 3
**Confidence:** 4

**Review:**

This paper presents the OLoRA+ algorithmic technology and a new hybrid method, PEFT, which combines OLoRA's orthonormal initialization. The idea behind the OLoRA mechanism is to use orthonormal initialization of adaptation matrices via QR decomposition. This allows for faster convergence and performance improvements compared to standard LoRA. This research demonstrates that the optimal optimization strategy fundamentally depends on the initialization scheme, offering a new dimension for hyperparameter tuning.
Advantages of paper:
- This paper demonstrates that the optimal optimization strategy fundamentally depends
on the initialization scheme, offering a new dimension for hyperparameter tuning.
- An optimal optimization strategy is demonstrated that fundamentally depends
on the initialization scheme, offering a new dimension for hyperparameter tuning.
The proposed theoretical model is demonstrated using a practical simulation, the results of which are shown in Tables 1-3.
Disadvantages of paper:
– an optimization problem that provides a solution to the stated problem is not formulated, and the proposed expressions (4)–(6) do not demonstrate a "synergistic" effect that reflects the possible results;
– the use of references to non-scientific literature, particularly Wikipedia resources, is questionable;
- it is not entirely clear how and where these results can be applied and what the methodology for their application will be.
In the reviewer's opinion, the proposed results are close to engineering modeling, not scientific research. The poor mathematical framework and the lack of a clear problem statement prevent a clear statement of the study's scientific contribution.

---

### Official Review · Reviewer_2Fht · 2026-03-12
**The Illusion of Synergy: A Conceptually Strong but Empirically Weak Hybrid**

**Rating:** 4
**Confidence:** 4

**Review:**

This paper introduces OLoRA+, a novel method that "synergistically combines the structural stability of Orthonormal Low-Rank Adaptation (OLoRA) with the accelerated convergence of LoRA+". The authors hypothesize that integrating structured adapter initialization with differential optimization yields a superior Parameter-Efficient Fine-Tuning (PEFT) framework.
Strengths The core idea of merging two complementary PEFT techniques is highly logical and possesses significant research potential. A major conceptual contribution is the discovery of two distinct learning dynamics: a "Refinement" strategy when the learning rate ratio is less than 1, and an "Exploration" strategy when it is greater than 1. The authors rightly point out that "the optimal optimization strategy is contingent on the initialization method". Furthermore, experiments on TinyLlama, OPT, and Gemma models demonstrate that the proposed approach improves upon the baseline "without incurring additional computational costs".
Weaknesses Despite the appealing hypothesis, the empirical foundation of this manuscript is critically weak and insufficient for proper validation. The models were fine-tuned on a drastically reduced subset of only 1000 training examples for a single epoch, which does not accurately reflect real-world LLM adaptation scenarios. Crucially, the experiments completely omit direct comparisons with the original LoRA and LoRA+ methods, even though the authors boldly claim that OLoRA+ "surpasses its predecessors". Furthermore, evaluating instruction-following capabilities using archaic n-gram metrics like BLEU and ROUGE fails to capture the actual semantic quality of the generated text. The study also lacks multiple runs with different random seeds, making it impossible to assess the statistical significance of the marginal metric improvements. Finally, the assertion that the paper provides a "deeper understanding of the interplay between adapter initialization and optimization" is not supported by rigorous theoretical or mathematical analysis.
Conclusion In its current state, this manuscript represents an unfinished draft rather than a complete and rigorous scientific study. To meet acceptance standards, the authors must conduct comprehensive training on full-scale datasets and include direct empirical comparisons with all predecessor methods mentioned in the literature review. Because the experimental pipeline requires such radical revisions, the paper is currently recommended for rejection.

---

### Decision · Program_Chairs · 2026-03-14

**Decision:**

Accept (Oral)

**Comment:**

Dear Author(s),

On behalf of the Program Committee of the International Conference on Mathematics of Artificial Intelligence (MathAI 2026), we are pleased to inform you that your paper has been accepted for an oral presentation at MathAI 2026.

Your paper was evaluated through a rigorous two-stage review process involving both automated screening and expert review by members of the Program Committee. The reviewers recognized the quality and contribution of your work.

Presentation details:

- Format: Oral presentation (15–20 minutes + 5 minutes Q&A)
- Mode: You may present either in person (offline) at the conference venue in Sirius, Russia, or remotely via Zoom. Please indicate your preferred mode when confirming your participation.
- Conference dates: Marh 30 - April 3, 2026
- Website: https://mathai.club

Next steps:

1. Please confirm your participation and presentation mode by replying to this email mathai.club@yandex.ru no later than March 15, 2026 18:00 Moscow time.
2. If you plan to attend in person, the organizing committee will provide accommodation details separately.
3. Please prepare your final camera-ready manuscript according to the formatting guidelines available at https://mathai.club and upload it to OpenReview by March 15, 2026 18:00 Moscow time.

Should you have any questions regarding the program, logistics, or your presentation slot, please do not hesitate to contact us.

We look forward to your contribution to MathAI 2026.

With kind regards,

MathAI 2026 Program Committee
International Conference on Mathematics of Artificial Intelligence
https://mathai.club
OpenReview: https://openreview.net/group?id=mathai.club/MathAI/2026/Conference
Telegram: https://t.me/MathAI_club
Email: mathai.club@yandex.ru